# “The Devices Themselves Aren’t the Problem”—Views of Patients and Their Relatives on Medical Technical Aid Supply in Home Mechanical Ventilation: An Explorative Qualitative Study

**DOI:** 10.3390/healthcare10081466

**Published:** 2022-08-04

**Authors:** Michael Ewers, Yvonne Lehmann

**Affiliations:** Institute of Health and Nursing Science, Charité—Universitätsmedizin Berlin, Corporate member of Freie Universität Berlin, Humboldt Universtität zu Berlin, 13353 Berlin, Germany

**Keywords:** home mechanical ventilation, technical aid supply, patient views, patient education, qualitative study, home care

## Abstract

(1) The supply of medical technical aids and the instructions on using them is critical for home-mechanically ventilated patients and their relatives. However, limited evidence exists on the needs-based nature of this care. (2) Aim: To gain insights into users’ views on this form of care, to identify key challenges, and to derive empirically sound preliminary recommendations for its future design. (3) Methods: An explorative qualitative interview study was performed in Germany. Semi-structured interviews were conducted with home-mechanically ventilated patients and their relatives. Patients were selected through purposive sampling. Interviews were audio-recorded, transcribed, and analysed using a content analysis approach. (3) Results: 27 patients and 9 relatives were interviewed. From their point of view, ventilation-specific equipment is generally reliable and is seen as a belonging of the patient. However, if the patient lacks competence in using the technology or if information or instructions are neglected, ambiguous and unsafe situations easily arise. (4) Conclusions: The present study is one of the first to provide insights into technical aid supply in home-mechanical ventilation from the users’ point of view. It highlights the need for continuous professional support and for evidence-based educational strategies that promote safety among the users of technical aids in home care.

## 1. Introduction

Internationally, the number of people of all ages who need various medical devices and technology-intensive forms of home care due to a wide range of health problems is on the rise. This rise is a consequence of demographic and epidemiological change, new industrial and market developments (technology push), changing social expectations and attitudes towards living with serious diseases and disabilities (technology pull), and the increase both in scientific and technological knowledge and in diagnostic and therapeutic possibilities [1,2]. Following the WHO, the term “medical device” is used in this context to refer both to “(…) any instrument, apparatus, implement, machine, appliance, implant, reagent for in vitro use, software, material or other similar or related article, intended by the manufacturer to be used, alone or in combination, for human beings” [3] and to corresponding accessories to these devices (e.g., connectors, filters, tubes, masks, and cannulas). These medical devices are used to support physiological functions, to apply and administer drugs or other substances (e.g., artificial nutrition), or to monitor vital parameters. The number of these medical technical aids is immense, but no uniform statistics yet exist that could provide information about their use. However, based on a recent systematic review [4], most of the research on these aids in home care focuses on respiration therapy in the form of long-term oxygen therapy or home mechanical ventilation (henceforth, HMV).

The exact number of people with HMV remains unclear. Only prevalence estimates exist from individual countries or regions, such as Poland [5], Hungary [6], Spain [7], the Netherlands [8], Germany [9], South Korea [10], and North America [11,12,13,14]. Although the indications and manifestations of HMV are extremely heterogeneous, all available studies confirm a general increase in this form of technology-intensive home care in many countries, with noninvasive HMV growing more significantly than the invasive form [14].

Despite this increasing use of HMV, limited research has been conducted on safety issues in technology-intensive home care, on the supportive care needs of patients and their families, and especially on the educational requirements for using technical medical aids, which are often critically important to sustaining the life of the user. Most research on patient safety has focused on the hospital setting, on objective safety parameters, on technical aspects, or on the perspectives of experts. However, some studies and reviews have also dealt with home care and the subjective dimensions of patient safety in this setting [15,16,17]. This growing body of literature has concluded that the minimally structured and regulated care environment, the greater participation of patients and relatives in everyday care tasks, the qualification of the involved providers, and regulatory issues all have a great impact on the safety of home care in general [18,19]. These challenges also arise in technology-intensive home care. Moreover, in terms of the perception of safety in this form of care, research on HMV has emphasised the critical role of multi-dimensional needs-based care, of building trusting relationships with professional care providers (especially nurses), and of providing ongoing information and instruction [20,21,22].

To our knowledge, no study has yet investigated in detail the provision of medical technical aids in HMV, the manner in which patients and relatives wish to be instructed in the use of the devices and supported in this use, or the effect that this instruction may have on their feeling of safety. This research gap is noteworthy because the supply of technical medical aids makes this form of home care possible in the first place, and the correct use of these aids can have a considerable influence on treatment outcomes, on the quality of life, and—not least—on the safety of the patients and their caregivers [14,23]. In addition, numerous different service providers are involved in supplying technical aids in HMV, which makes this form of care even more complex. In Germany, for example, authorised providers of medical devices and equipment are responsible for the timely provision, control, maintenance, and technical emergency assistance of medical technical aids, as well as for instructing and counselling patients or their caregivers on using this equipment in home care settings, be it in private homes or in assisted living communities (henceforth, ALCs). Exactly how patients or their caregivers carry out these tasks, the knowledge that they have in using the equipment, and—in particular—how needs-based these services are remain unclear. Furthermore, little is known about the interaction that patients or their caregivers have with other service providers, such as staff from in-home nursing care services or general physicians. Therefore, for the present qualitative research study, we investigated how mechanically ventilated patients and their relatives perceive the supply of medical technical aids in HMV and how the supportive care and educational needs of the patients and their relatives are met by the involved care providers.

The aim of this research was to gain insights into users’ views on technical aid supply in HMV, to identify key challenges from their perspective, and to derive empirically sound preliminary recommendations for the future design of this form of technology-intensive home care, particularly regarding safety aspects. This investigation is part of a large-scale, multi-phase health services research investigation into the “Safety dimensions of aid supply in homecare-ventilated patients” (SAVENT), which was conducted in Germany from 2019 to 2022.

## 2. Materials and Methods

### 2.1. Study Design

An explorative qualitative cross-sectional study based on semi-structured interviews with mechanically ventilated patients and their relatives was conducted. The methodology and reporting of this study followed the Consolidated Criteria for Reporting Qualitative Research (COREQ) [24].

### 2.2. Setting, Recruitment, and Participants

The study was conducted in Germany. A purposive sampling technique was used to recruit study participants with a maximum variation in characteristics, for example, in terms of age, gender, underlying diseases, form of ventilation (invasive vs. noninvasive), extent of skilled nursing care, and living condition (alone or with relatives; in a private household or in an ALC; in a rural or urban area). Nurses who had daily contact with the families were used as gatekeepers. They provided basic information orally and distributed an introductory letter about the study to eligible participants. In addition, caregiver groups and German self-help organisations (e.g., for COPD patients) were used for recruitment. Patients or relatives interested in participating were referred through the home care providers or contacted the research team directly to schedule an appointment for an interview. All study participants had to be at least 18 years old, to communicate in and understand German, and to provide written consent in order to participate in the study.

### 2.3. Data Collection

The data for this study were collected between July 2019 and March 2020 in different regions of Germany. Interviews were conducted face-to-face in private homes or in ALCs. On average, they lasted 45 min (range: 14–105 min). All interviews were digitally recorded. Creative approaches were taken to not exclude patients with ventilator-related communication problems from the study [25]. In one case, an eye-driven tablet communication system was used (Eyegaze Edge^®^), which caused the interview to last 230 min (210 of which were recorded). In another case, for the same reason, an email interview was conducted. If necessary, relatives were included in the patient interviews in a supportive manner; in two cases, they were interviewed on a proxy basis. If relatives were involved, they were also asked about their own views on the home care situation.

The interviews followed a flexible guideline that covered the following five topics: (a) the ventilation itself and the current living conditions, (2) first experiences with the home care situation, (3) the relevance of ventilation-associated technical aids from the patient’s (or relatives’) point of view, (4) dealing with limited communication abilities in home care, and (5) safety from the patient’s and relatives’ point of view as well as strategies for maintaining it. New questions evolved during data analysis, and topics became more focused in later interviews. In all cases, two researchers were present during the interviews. While one researcher conducted the interview, the other took notes on the residential environment, the course of the interview, possible disturbances, and topics that were addressed before and after the interview audio was recorded. These notes were documented in an interview protocol together with socio-demographic, disease-specific, treatment-related, and other relevant information.

### 2.4. Data Analysis

All electronically recorded audio data were transcribed verbatim and pseudonymised. Together with the interview protocols, the transcripts were imported into qualitative data analysis software (MAXQDA Version 18 by VERBI Software GmbH; Berlin, Germany). All data were iteratively reviewed by at least two researchers and analysed using a content analysis approach [26]. Main categories were defined a priori, and sub-categories that take particular account of the context of each statement were developed based on the data. The development of the main categories was sensitised by the research goals and questions, by the German legal framework of aid supply, and by conceptual considerations of patient safety and quality in technology-intensive home care [27,28]. In an iterative process, comparisons were made between and within interviews in order either to aid in formulating more in-depth questions or to guide the pragmatic reconstruction of the manifest and latent content of the interviews. To assure transparency and intersubjective comprehensibility, codes, categories, and their interpretation were recursively validated against the material [29]. Analyses were discussed, condensed, and revised within the research team, and primary focus was placed on developing main categories.

## 3. Results

The study sample comprised 27 patients and 9 relatives. Twenty patients were able to complete their interviews independently. At the patients’ request, seven interviews were conducted together with a family member. In two cases, relatives were interviewed alone as proxies. Table 1 provides information on the demographic characteristics of the patients included in this study either directly (as interviewees) or indirectly (via their relatives), and whether the interview was conducted alone with the patient, together with a relative, or exclusively with the relative as a proxy is noted. Additional information relates to the housing situation, the main cause of ventilation, the type and daily duration of ventilation, the extent of skilled nursing care or personal care, and housekeeping assistance provided by an in-home nursing care service (on a 24/7 basis or intermittently from 1 to 3 times per day). The sociodemographic data of the relatives included in the interview study are summarised in Table 2.

A total of 29 interviews were analysed as described above. Interview analysis revealed four major categories, with two sub-categories each (see Table 3). The results are presented along the outline of the four main categories, with selected quotes for each main category.

### 3.1. The Journey of Ventilation Use Begins

In Germany, according to the medical guideline for “non-invasive and invasive ventilation as therapy for chronic respiratory insufficiency” [23], patients’ and relatives’ journeys with both noninvasive and invasive mechanical ventilation regularly begin in the hospital. This may be the result either of an ad hoc deterioration (acute initiation) of the patient’s condition or of a slowly progressive deterioration of the patient’s health status (elective initiation). In the latter case, some respondents found mechanical ventilation to be a relief. When symptoms decreased and the quality of life improved noticeably, patients had the feeling of “breathing a sigh of relief”. Most respondents, however, described the initiation of the ventilation as a severe break from normal life and an exceptional existential situation for both themselves and their relatives. Only gradually did the patients realise the consequences of this therapeutic intervention and their accompanying dependency on life-sustaining technology. It was difficult for patients to get used to the devices attached to their bodies and to the associated nursing care procedures, and they usually faced all of this with great anxiety. As a result, patients had no choice but to fully comply with all of the instructions provided by the medical professionals at the hospital, some of whom tried to motivate the patients and their family members to begin self-care as soon as possible.


*“At first, the nurses took over everything. And then, all of a sudden, they told me, ‘Try doing it yourself.’ And yeah, that didn’t work for me at first. Not at all. I was, like, blocked inside. [...] They talked me into doing it, and then, step by step, I started to do it, but only if someone was at my side.”*
(P20)

At this stage, respondents were overwhelmed by the technology and all that it entailed. They found themselves easily distracted (e.g., by the pressure of the mask, by the tubes and cannulas, or by the sounds of the ventilator) and—in many cases—unfocused, not very receptive, and unable to learn. The respondents often could not recall whether they had received any instruction from nurses, doctors, or staff from a medical device and equipment supplier during their hospital stay or—if so—what exactly had been explained or demonstrated. Some said that they thought from the very beginning that they would not be able to remember technical details or procedures at all, whereas others assumed that their dependency on the medical devices was temporary, and they thus did not pay full attention to the accompanying instructions or to their own learning needs. If health professionals made any effort to explain to patients or their relatives the situation, the principles of ventilation, the equipment used, or even simple manual procedures of self-care, these efforts had little sustainable effect.

### 3.2. A Bumpy Start, but Onwards We Go

After being discharged to their homes or to an ALC, the situation changed abruptly for patients and their relatives. Although they received some support from a medical device and equipment supplier and an in-home nursing care service, they experienced various initial problems while trying to adjust to the ventilation, the technology, and life at home. On the one hand, problems resulted from the new living conditions, the continuing need for personal and skilled care, and the imperative to develop new daily routines. On the other hand, respondents repeatedly encountered problems in handling the technical equipment. Initially, the medical technology was perceived as a foreign element in the home environment. In addition, family members were overwhelmed by the maintenance and monitoring tasks assigned to them. Some respondents reported having had the impression that the technology did not work properly. Alarms went off, and external rechargeable batteries suddenly came off. When the patients called for help, operating errors often led to a malfunction. Such experiences greatly unsettled some of respondents.


*“I drove everyone crazy because I was a bit unsure myself. And then you make a huge fuss […]. But that’s also logical if something doesn’t work over and over again… you wonder what’s actually working and what isn’t.”*
(P15)

In such nerve-wracking situations, the various service providers involved in HMV were of limited help from the respondents’ point of view. In fact, the legally required instruction that comes with the devices provided by the staff of the medical device and equipment supplier varied from person to person, focused mostly on technical elements, and was often used only episodically and on an ad hoc basis. The staff of in-home nursing care services—who were more frequently present and thus available more easily—often seemed to have little familiarity with the medical technology that the patient used in each case. Occasionally, they appeared to be as helpless to solve the problems with the medical technical devices as the patients or relatives themselves, which did not necessarily inspire confidence in the competence of the in-home nursing care services. Patients and relatives consistently reported that despite these eye-opening experiences, they usually had no choice but to come to terms with the circumstances and to get used to living with a technical dependency and the recurring feelings of anxiety and danger.

### 3.3. The Complex Daily Routine with HMV

Some respondents reported that as time progressed, they began to get a handle on the technical element of their home care. By learning that alarms, mistakes, or malfunctions were often the result of operating errors, the need to replace consumables, or over-sensitive alarm systems rather than of broken equipment, confidence in the medical devices gradually grew. As their understanding of the technological aspect of the care evolved, respondents sometimes even recognised ways to make their lives easier with the aids. Some respondents—mostly relatives—began to investigate possible innovations in the medical technology market. They looked for ventilators, masks, tracheal cannulas, or tubing systems that were better designed and tailored to the needs of the often-elderly users, some of whom had fine motor impairments. The costs of this user-friendly equipment were not always covered by the health insurance companies, and some respondents thus purchased the equipment themselves. A few respondents developed greater competence in this gradual process of adapting to the technological aids than others, who continued to rely heavily on third-party support. Moreover, patients’ acceptance of their permanent dependency on medical devices did not grow with increasing technical understanding and skills in all cases.


*“The whole technical aspect… that’s obvious, depending on how technically skilled you are. But the adjustment [you have to make] to living with something like that… that probably needs a bit more support from someone, yeah.”*
(P11)

Instead of being supported in this demanding coping process, patients and relatives were often confronted with multiple care management tasks that had to be performed in order to keep their complex daily routines running as smoothly as possible. One of the key challenges was ensuring a reliable supply of medical technical devices and consumables. Although this was the contractual responsibility of the medical device and equipment supplier, at times, the supplier delivered incorrect or inappropriate equipment. The large number and variety of materials used in HMV was a major challenge for all parties involved. Patients and relatives were forced to carefully check each delivery to avoid ending up with the wrong aids. Even minor events—such as replacing an assistive medical device due to a healthcare insurance’s cost-saving initiatives, or the delivery of an unsuitable cannula or a missing component for the ventilation—had the potential to disrupt care routines that had been tediously developed. Moreover, respondents frequently reported having felt anxious and unsafe—a feeling that was often difficult to control.

### 3.4. The Struggle to Stay Safe

Respondents often felt that they had to be on constant alert. This was due—on the one hand—to the error-prone supply processes or because the maintenance and cleaning processes of the aids were not adapted to the home environment. On the other hand, the feeling of needing to be on constant alert was also due to the fact that the professional care providers did not always provide the attention to detail that patients or family members expected from them, which sometimes nearly led to accidents in HMV. Some of the respondents then complained but received inadequate responses. In fact, they often felt that they were seen as being overly demanding and would be ignored moving forwards. Others tried to deal with risks, incidents, and complications on their own. One older couple reported that the patient had only been able to sleep on his left side for three years because the ventilator tube was too short. Once, the patient had accidentally turned over in his sleep and had pulled down the entire machine with him. The care providers did not notice this burdensome situation and did not offer any help that could have improved the couple’s situation. The couple believed that there was no other solution and simply accepted the situation. Other respondents also did not want to burden anyone and, therefore, for example, used disposable consumables more than once and repaired wobbly electronic contacts or batteries with bandages. In so doing, these individuals took—consciously or unconsciously—considerable safety risks.


*“And sometimes, I don’t know whether I should take the time to ask one question or another, or whether I should instead try to solve the problem myself—basically, learning by doing. Besides, the others manage to figure it out [somehow].”*
(R1)

This “tinkering around” did not always end well, and some respondents reported accidents, including pneumonia caused by incorrectly performed suctioning. As the staff of the medical device and equipment suppliers often turned their attention to the next patient admitted to HMV, those already receiving home care were usually left alone with many unanswered questions that arose over time. This was especially true for patients with noninvasive ventilation and for those without relatives. Some were able to gradually acquire minimal expertise in using the medical aids and in HMV in general, thereby also gaining back autonomy and self-confidence. Nevertheless, new questions and challenges continued to arise over time. In some cases, there was a desire to better understand how the life-support devices worked and how they could be used to make life easier. In such situations, some respondents searched for information on the internet or looked for people in similar situations to exchange experiences and knowledge. Self-help communities were perceived as a valuable yet only partially reliable source of information and support. The expectation that the advantages and disadvantages of certain aids or procedures would be explained or that greater support would be provided that would better enable individuals to care and be responsible for themselves was rarely met. Only when they pushed very hard or asked intensively for such assistance were these individuals given professional educational support. However, this was not an option for all respondents, especially for vulnerable patients in poorer health or for those with less education, those with higher levels of anxiety, or those who had no relatives or social support.

## 4. Discussion

The objective of the present study was to explore the practice of medical technical aid supply and its associated challenges from the point of view of patients in HMV and their family members in Germany. Four main categories and eight subcategories were identified in the data that captured patients’ and relatives’ views of the supply and handling of medical technical devices, with a particular focus on the feeling of safety. The findings of this exploration—discussed here in comparison with existing evidence on the topic—should help in identifying the specific needs of supportive care and the education associated with this form of technology-intensive home care. The findings may also be relevant in other countries and in various other circumstances.

Our results reveal that technical medical aids—which initially appear to be the most important component of HMV—are rarely a problem upon closer inspection. Real mechanical or electronic malfunctions—such as insufficient ventilator performance (e.g., in-/expiration pressure) or software-related alarm failures—are rare [30,31], and the technical quality of the devices is not questioned by suppliers or users. Patients and relatives find the devices reliable and trust them, even if only after a certain familiarisation period. Gradually, ventilators even come to be seen as a functioning and self-evident extension of the person to whose body they are directly connected. This process of stepwise assimilation has been described by other authors, e.g., [22,32,33]. Moreover, some evidence suggests that users occasionally want slightly easier-to-use aids that are designed to appeal more to their specific limitations (e.g., relating to sensory or fine motor issues) or that they feel provide more comfort (e.g., masks) [33]. Even though these devices are available on the market, patients and relatives may not always have access to them (e.g., due to decisions made by health insurers or by medical device and equipment suppliers). However, consistent with existing evidence [34,35,36] (including in the data collected here), the real challenges in technology-intensive home care are more likely caused by human factors, such as adjustment, maintenance, or handling problems.

What patients and relatives experience de facto is a lack of thoughtful, timely, reliable support when it comes to dealing with technical medical aids. Notable examples include reported deficiencies in maintaining ventilators, which are necessary to control the technical risks of HMV. Standards of maintenance intervals and procedures as well as the communication and exchange of device-related information between the involved healthcare staff (e.g., about malfunctions, changes in device performance, or ventilator settings) often seem to be poorly established, as other researchers have stated [14,31]. Risks may also emerge from the often selectively performed therapy check-ups [30]. From the user’s perspective, significant room for improvement exists in terms of aid suppliers’ diligence in delivering replacement components or consumables and in coordinating with nurses or other healthcare workers in the home care setting. The high level of monitoring and coordination that patients and their families believe they must provide in terms of aid supply in the interest of their own safety is a direct reaction to this lacking diligence on the part of the suppliers [30,37]. This finding draws attention to the role of external commercial medical device and equipment suppliers in HMV and to their legally mandated safety responsibilities in adjusting, maintaining, and applying these devices. Earlier European studies on the topic concluded that collaboration between external suppliers, prescribers, and users of medical technical aids in technology-intensive home care was low, as was participation in joint quality control initiatives [30,36]. This aspect was also evident in the research conducted here. In fact, to our knowledge, no independent evaluations have yielded any evidence that suppliers meet patient and family needs or common quality and outcome parameters. The apparent neglect in continually educating the users of medical technical aids is the greatest safety risk in technology-intensive home care, especially as this neglect often results in application errors and adverse events [4,14]. This applies to all parties involved, including patients and their relatives as well as the staff of in-home nursing care services and sometimes also the prescribing doctors.

The use of medical technology and ”ompl’x life-sustaining therapies in patients’ private living environments is no longer exceptional in many countries. This technology provides significant benefits to patients, including improvements in quality of life and cost savings [22], but it is not without risks [23,37]. Although no consistent incident reporting system exists for capturing the improper or inattentive use of medical devices and equipment, problems such as difficulties in the physical setting in which the care is provided, infections, accidents, oxygen toxicity, and other adverse technology-related events in HMV can be assumed to occur frequently [28,30,31]. Users may sometimes unintentionally cause these problems themselves when carelessly handling aids (e.g., turning off noisy alarms) or when their questions go unanswered, thereby causing the users to feel pressured into using trial and error. Issues related to device or interface settings (e.g., tolerance of ventilation pressure, pressure sores, leakages, and symptom control) may then be detected too late or not at all [15,21]. The literature has already revealed that knowledge about safe ventilator use in HMV is particularly low [4] and that inadequate user training leads to uncertainties and to the misuse of aids, with a potentially adverse impact on health [14,30,31]. This evidence is strongly supported by the findings of our qualitative study and should therefore be reviewed carefully in conceptual adjustments in medical-technical aid provision in HMV.

Research on populations with complex needs in home care has revealed that information and educational support are important yet unmet needs [21,36,37,38,39], as was also found in the accounts of the interviewees in the present study. A pressing need exists for the proactive informing and counselling of patients and relatives; for flexible, targeted, and continuous instruction; for user involvement in care and decision-making, particularly in terms of the use of technology [34,36,39]. During our analysis, it became evident that education should always be adapted to the actual situation of the patients and their relatives and should take into account their specific support and learning needs and their capabilities throughout the trajectory from hospital to home-based care to living with the dependency. Moreover, education should not be limited to technical aspects or application issues; instead, it is critical to pay attention to the personal coping and adaption processes of the users and the challenges they face in living at home with a disease and with technology [22,33,38]. The balance between the patient’s and their family members’ desire to retain or regain control and autonomy on the one hand and the need for active support on the other hand should be carefully maintained [21,39].

The demanding task of continuously supporting and educating patients and relatives cannot be left to medical device and equipment suppliers alone; rather, it is a team effort that must involve not only the staff of the discharging hospital, but also the staff of the in-home nursing care service and—especially—the prescribing physicians [40]. However, as nurses work in hospitals for medical device and equipment suppliers and for in-home nursing care services (and sometimes also in doctors’ offices), they play an essential role in ensuring that ongoing support and education are provided in a way that meets the needs of the technology-dependent patients and their relatives [23,41]. This, of course, requires that nurses have sufficient competencies not only in handling medical technical devices and equipment, but also in dealing with underlying diseases, illness trajectories, and coping processes of patients or their relatives as well as in educational concepts and approaches. Ensuring safety in technology-intensive home care—both objectively and subjectively—should be nurses’ top priority and professional imperative. Therefore, apart from educating patients, it is also important to educate healthcare professionals—and especially nurses—on the different aspects of medical technical aid supply and of living with life-sustaining technology in private homes or in ALCs.

### 4.1. Preliminary Recommendations

At this point, first recommendations for the future design of medical technical aids supply in HMV are formulated based on the available data, their analysis, and discussion. However, these recommendations are still preliminary (see Table 4). They will have to be supplemented by other results of the broader SAVENT project and thus also by additional perspectives (such as those of service providers). Finally, they need to be contextualised, specified, and addressed to different stakeholders and decision-makers.

### 4.2. Limitations

The present research has some limitations that should be considered when applying its findings. Research took place in HMV in Germany and thus under circumstances that may differ from those in other countries. This applies to the legal framework for the provision of medical technical aids, to the role of medical device and equipment suppliers, to these suppliers’ specific responsibilities, and to the suppliers’ collaboration with other service providers in home care. The role and competencies of nurses in technology-intensive home care and in in-home nursing care services may also differ due to differences in educational standards. Therefore, not all findings are transferable to contexts in which the role of these providers is different or in which mechanical ventilation is mostly performed under different legal, institutional, or qualification circumstances.

It should also be noted that some of the interviews in this qualitative study were conducted with patients alone, whereas others were held with patients and relatives together, and still others were conducted only with relatives. This procedure is unusual in qualitative research as it is difficult to explore possible disparities in the interviewees’ points of view in this way. In addition, some of the interviews used technical communication aids, which may have produced a different type of data than is usual in verbal interviews. However, we opted for this procedure in order to also be able to include the most vulnerable patients and their experiences, as argued in the literature e.g., [42,43]. Where divergent views between patients and relatives existed or the data deviated from those of the other interviews, these differences were carefully considered in the analysis. In order to minimise bias in interpreting the data, we used a team approach to the analysis.

Although we paid much attention to purposefully recruiting study participants, not all characteristics are sufficiently represented (e.g., ethnic diversity). Other factors—such as respondents’ involvement in support groups or patient organisations—may be overrepresented due to the recruitment strategy. Overall, the sample captures a wide range of different backgrounds, living and care situations, care experiences, and personal views, and it thus forms a solid basis for answering the research question. The size of the sample, its multidimensional nature, and the depth of interviews conducted are sufficient to ensure data saturation [44,45]. 

The data in this study were collected shortly before the outbreak of the COVID-19 pandemic. At that time, telehealth services for either monitoring, counselling, or patient education were used in Germany in a few demonstration projects at best. In the meantime, the potentials of telehealth systems have been more widely recognised [46]. This aspect should therefore receive more attention in future studies on HMV.

Exploring the views of the staff of medical device and equipment suppliers and of in-home nursing care services was not part of this study, and no conclusions can be reached on the topic. However, this important aspect was investigated in another study in the more comprehensive SAVENT project mentioned in the background. Independent publications on this topic are expected in due course. This will then allow a multi-perspective view on medical technical aid supply in technology-intensive home care.

Nevertheless, our findings have important implications for the future design of medical technical aid supply in HMV and for the development and evaluation of needs-based concepts that support and educate patients and relatives in this context. These implications relate primarily to the dimensions of safety in HMV and specifically to technical and objective safety as well as—more importantly—to the feeling of safety among patients and their family members.

## 5. Conclusions

To our knowledge, this is one of the first studies to explore the supply of medical technical devices and aids in technology-intensive home care in depth from the point of view of the patients and their relatives. Our findings are particularly useful for designers of medical technical devices, for home care providers, and for nurses and other healthcare professionals who work in this field because they should aid in better understanding the needs of patients and their relatives and in ensuring patient safety in medical aid supply in technology-intensive care (e.g., mechanical ventilation, haemodialysis, infusion therapy, and total parenteral nutrition). As this form of healthcare shifts from the hospital to the home environment, it is becoming increasingly important to improve the usability and intuitive handling of technical medical devices and equipment for use in less-controlled home environments. Furthermore, this study highlights the direction of ongoing supportive care and education with the goal of promoting users’ sense of safety in this environment, thereby also contributing to research on patient safety in healthcare at home—a previously understudied area of healthcare. Finally, the study emphasises the need for evidence-based concepts of supportive care and education in medical technical aid supply and stresses that the involved healthcare professionals—especially nurses in all settings—must be better prepared for their demanding tasks in technology-intensive home care.

## Figures and Tables

**Table 1 healthcare-10-01466-t001:** Patients’ sociodemographic, disease-specific, and treatment-related characteristics.

Code	Sex	Age	Housing *	Main Reason for HMV	Type of Ventilation	Vent. Hours per Day **	Home Nursing Care	Interviewed Person
P1	F	65	ALC	Infection, multimorbidity	Invasive	<16 h	24/7	Patient and relative
P2	M	53	Private home	Neuromuscular	Invasive	24 h	24/7	Patient and relative
P3	M	30	Private home	Neuromuscular	Noninvasive	<16 h	24/7	Patient
P4	M	54	ALC	Neuromuscular, COPD	Invasive	24 h	24/7	Patient
P5	F	59	Private home	Neuromuscular	Invasive	24 h	24/7	Patient
P6	M	60	ALC	COPD	Invasive	<16 h	24/7	Patient and relative
P7	F	70	ALC	COPD	Invasive	<16 h	24/7	Patient
P8	F	75	ALC	Neuromuscular	Invasive	>16 h	24/7	Patient and relative
P9	M	75	Private home	COPD, post-polio	Noninvasive	>16 h	Intermittent care	Patient and relative
P10	M	79	ALC	COPD	Invasive	>16 h	24/7	Patient
P11	M	75	Private home	COPD	Noninvasive	<16 h	Intermittent care	Patient
P12	F	69	ALC	Neuromuscular, multimorbidity	Invasive	<16 h	24/7	Patient
P13	M	70	ALC	COPD	Invasive	24 h	24/7	Patient and relative
P14	M	58	ALC	COPD	Invasive	<16 h	24/7	Patient
P15	F	74	Private home	COPD	Noninvasive	<16 h	Intermittent care	Patient
P16	M	69	ALC	COPD	Invasive	24 h	24/7	Patient
P17	M	69	ALC	COPD	Noninvasive	<16 h	24/7	Patient
P18	M	69	Private home	COPD	Noninvasive	<16 h	Intermittent care	Patient
P19	M	70	ALC	COPD	Invasive	<16 h	24/7	Patient
P20	F	70	ALC	COPD	Invasive	>16 h	24/7	Patient
P21	F	69	ALC	COPD	Invasive	>16 h	24/7	Patient
P22	F	71	ALC	COPD	Noninvasive	<16 h	24/7	Patient
P23	M	58	Private home	Neuromuscular	Invasive	24 h	Intermittent care	Patient
P24	M	75	ALC	COPD	Invasive	<16 h	24/7	Patient
P25	F	75	ALC	Neuromuscular	Invasive	24 h	24/7	Relative (as proxy)
P26	F	73	ALC	COPD	Invasive	24 h	24/7	Relative (as proxy)
P27	F	64	Private home	COPD	Noninvasive	<16 h	Intermittent care	Patient
P28	F	71	Private home	COPD	Noninvasive	>16 h	Intermittent care	Patient
P29	M	65	ALC	Infection, multimorbidity	Invasive	24 h	24/7	Patient and relative

** ALC (Assisted living community). * At the time of the interviews, mechanical ventilation had been being used from 4 to 30 months.

**Table 2 healthcare-10-01466-t002:** Relatives’ sociodemographic characteristics.

Code	Sex	Age	Living Together with Patient	Relation to Patient	Employment Status	Type of Interview
R1	M	67	No	Husband	Retired	Relative and patient
R2	M	44	Yes	Spouse	Employed	Relative and patient
R6	F	58	No	Wife	Unemployed	Relative and patient
R8	M	67	No	Husband	Employed	Relative and patient
R9	F	65	Yes	Wife	Retired	Relative and patient
R13	F	64	No	Wife	Employed	Relative and patient
R25	F	31	No	Daughter	Employed	Relative (as proxy)
R26	F	65	No	Wife	Retired	Relative (as proxy)
R29	F	53	No	Wife	Employed	Relative and patient

**Table 3 healthcare-10-01466-t003:** Four main categories and eight subcategories.

Main Categories	Subcategories
The journey of ventilation use begins	▪ Being in an exceptional existential situation ▪ Being dependent, distracted, and unable to learn
A bumpy start, but onwards we go	▪ Trying to adjust to ventilation, technology, and life at home ▪ Feeling challenged by collaboration with care providers
The complex daily routine with HMV	▪ Gradually building trust in the technology ▪ Feeling the need to be constantly on guard
The struggle to stay safe	▪ Dealing with risks, incidents, and complications ▪ Being left with many unanswered questions

**Table 4 healthcare-10-01466-t004:** Main problems and preliminary recommendations.

	Main Problems Identified in Medical Technical Aid Supply
#1	Technology is an enabling factor in technology-intensive home care and often considered to be the most important component of HMV. However, while the focus of all parties involved is often on the quality, functionality, and reliability of the medical technical devices and equipment, the main problems are more likely to be caused by human factors such as adjustment, maintenance, or handling problems.
#2	Patients and relatives de facto experience a lack of thoughtful, timely, reliable support when it comes to dealing with technical medical aids. The interactions with them are often episodic in nature, unilaterally focused on functionality, and not very empathetic about the coping demands patients and relatives are confronted with when living with technology-dependency in private homes or in ALCs.
#3	The knowledge about the safe use of medical technical aids is particularly low in HMV, as is health literacy in general. In combination with inadequate information, counselling, and instruction of users, this often leads to uncertainties and to the inappropriate use of technical devices, as well as to considerable safety risks. This affects patients, their relatives, as well as informal and formal caregivers in home care.
#4	A lack of collaboration between the several parties involved in HMV as well as a barely coordinated approach to medical technical aid supply can lead to specific safety risks and loss of quality. This is often exacerbated by insufficient qualifications of the healthcare professions involved, in terms of the technical, clinical, and personal dimensions of technology-intensive home care.
	**and preliminary recommendations directed towards them**
#1	Change the perspective away from the technology and towards the people who use it. Their complex daily tasks and challenges in using medical technology in their private home environment, as well as in managing the multidimensional safety risks in HMV or other forms of technology-intensive home care, should be at the very centre of a needs-based and person-centred medical technical aid supply.
#2	Make patients’ and relatives’ experience of care and their feeling of safety a prime indicator of service quality in medical technical aid supply. Provide continuous support, take a multidimensional approach, and strengthen the emotional coping of patients and relatives through social interaction to help them to regain and maintain their autonomy despite their dependency on medical technology.
#3	Provide medical technical aids together with needs-based information and instruction about their functionality, their proper use in the home care environment, and important safety precautions to reach expected outcomes. Develop and implement well-structured and evidence-based concepts for patient education in technology-intensive home care, using digital tools, and supporting media where appropriate.
#4	Conceptualise a needs-based medical technical aid supply as a team-effort and work diligently and collaboratively together with all parties involved (including patients and relatives). Improve the qualification of the professionals, especially of the nurses, because their technical, clinical, and social competencies have a direct impact on the users’ care experiences, safety, and quality of the technology-intensive home care.

## Data Availability

Not applicable.

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
