# Peer review of "“The Devices Themselves Aren’t the Problem”—Views of Patients and Their Relatives on Medical Technical Aid Supply in Home Mechanical Ventilation: An Explorative Qualitative Study"

_healthcare, 2022, doi:10.3390/healthcare10081466_

Round 1

Reviewer 1 Report

The authors of the article presented an original text dedicated to  views of patients and their relatives on medical technical aid supply in home mechanical ventilation.

The authors collected interviews of patients and their relatives. And they analyzed these interviews.

The methodology and reporting of this study followed the Consolidated Criteria for Reporting Qualitative Research (COREQ).

Despite the originality of the study, I believe that this article cannot be published in its current form. Here are my comments.

1. The objectives of the study and the conclusion do not correspond to each other. What kind of deep understanding of the views of users has been achieved?

2. What empirical sound recommendations have been formulated?

The conclusions of the article look extremely general and banal. It is obvious without research that qualified professionals and educational strategies are needed. Especially when it comes to patients aged 60+.

Author Response

Dear Reviewer 1,

Thank you for the peer review of our article. Please find our answers to your comments in the attachment.

Kind regards

Reviewer 2 Report

it is well written paper with unique subject but it need further study to survey doctors and nurses opinions on these results

Author Response

Dear Reviewer 2,

Thank you for the peer review of our article. Please find our answers to your comments in the attachment. 

Kind regards

Reviewer 3 Report

Since limited evidence exists on the needs-based nature of home mechanically ventilated patients, in this study the authors aimed to gain deeper insights into users’ views on this form of care and to derive empirically sound recommendations for its design. An explorative qualitative interview study was performed in Germany by a semi-structured interviews with home mechanically ventilated patients and their relatives. They studied 27 patients and 9 relatives. From their point of view, ventilation-specific equipment is generally reliable and is seen as a belonging of the patient. However, if the patient lacks competence in using the technology or if information or in-structions are neglected, ambiguous and unsafe situations easily arise. They concluded that their study is one of the first to provide insights into technical aid supply in home mechanical ven-tilation from the users' point of view highlighting the need for qualified professionals and for educational strategies that promote safety among the users of technical aids in home care.

The study is of interest providing an original approach. I have only a minor point to suggest. The authors should discuss the potential beneficial impact of telemedicine to improve the management of home mechanically ventilated patients, especially in the era of covid-related pandemia that significantly have reduced the access to non-covid patients to the hospital.

Author Response

Dear Reviewer 3,

Thank you for the peer review of our article. Please find our answers to your comments in the attachment. 

Kind regards
